# Enfoque integrado de NIRS, Machine Learning y programación genética para la estimación explicable del contenido proteico en cereales

**David Zarandieta Ortiz**
Dpto. Ingeniería en Sistemas Informáticos y Telemáticos.
Centro Universitario de Mérida, Universidad de Extremadura
davidzo@unex.es

**Francisco A. Galea-Gragera**
Área Pastos y Cultivos Forrajeros
Instituto de Investigaciones Agrarias
Finca La Orden-Valdesequera del
Centro de Investigaciones Científicas y Tecnológicas de Extremadura (CICYTEX)
francisco.galea@juntaex.es

**Francisco Chávez de la O**
Dpto. Ingeniería en Sistemas Informáticos y Telemáticos.
Centro Universitario de Merida, Universidad de Extremadura
fchavez@unex.es

**Fernando Llera Cid**
Área Pastos y Cultivos Forrajeros
Instituto de Investigaciones Agrarias
Finca La Orden-Valdesequera del
Centro de Investigaciones Científicas y Tecnológicas de Extremadura (CICYTEX)
fernando.llera@juntaex.es

**Josefa Díaz-Álvarez**
Dpto. Tecnología de los computadores y las comunicaciones
Centro Universitario de Mérida, Universidad de Extremadura
mjdiaz@unex.es

## Abstract

La cuantificación precisa del contenido proteico en cereales es esencial para optimizar su calidad nutricional y su valor agroindustrial. Sin embargo, los métodos tradicionales, como Kjeldahl y Dumas, presentan limitaciones en términos de costo, tiempo y destructividad de las muestras. En este estudio, se propone un enfoque basado en espectroscopía de infrarrojo cercano combinado con técnicas avanzadas de aprendizaje automático y programación genética para la predicción explicable del contenido proteico en muestras intactas de avena, cebada y triticale.

Se aplican tres algoritmos de Machine Learning (Regresión de Mínimos Cuadrados Parciales, Random Forest y k-Nearest Neighbors) para modelar la relación entre los espectros y los valores de proteína (valor predictivo), optimizando su rendimiento mediante GridSearchCV. Posteriormente, la programación genética permite generar

una expresión matemática interpretable que integra las predicciones de los modelos de ML, mejorando la precisión final del sistema.

Los resultados muestran que los modelos basados en preprocesamientos espectrales avanzados logran predicciones altamente precisas ($R^2 > 0.97$ en el mejor caso). La combinación de espectros de infrarrojo cercano, machine learning y programación genética demuestra ser una solución eficiente, no destructiva y escalable para la estimación del contenido proteico en cereales, con aplicaciones directas en la industria agroalimentaria y en la mejora de cultivos.

# 1  Introducción

Los cereales son cultivos esenciales en la agricultura global y constituyen una de las principales fuentes de nutrientes en la dieta humana y animal. Su composición incluye carbohidratos, vitaminas, minerales, lípidos y, especialmente, proteínas [1], cuya cantidad y calidad determinan su valor nutricional y comercial [2]. La proteína en los cereales desempeña un papel fundamental en la alimentación humana, influyendo en el crecimiento, la regulación metabólica y el sistema inmunológico [3]. El contenido de proteínas de los cereales difiere según las especies y presenta una gran variabilidad intraespecífica, como consecuencia de la interacción genotipo-ambiente, que caracteriza a este componente [4].

El contenido proteico de los cereales varía entre un $6\%$ y un $16\%$, dependiendo de la especie y las condiciones de cultivo. Las proteínas de los cereales se clasifican, atendiendo a sus características de solubilidad, en cuatro grupos: albúminas, globulinas, prolaminas y glutelinas. A medida que avanza la maduración del grano, la proporción en el mismo de las fracciones proteicas se modifica, aumentando con mayor rapidez las proteínas de reserva que son insolubles (prolaminas y glutelinas) que las citoplasmáticas (albúminas y globulinas). Estas variaciones afectan a la digestibilidad, al valor nutricional y las propiedades tecnofuncionales de los cereales, como la capacidad de formación de masas y la textura de los productos finales [5].

La creciente demanda de alternativas sostenibles a las proteínas de origen animal ha impulsado un renovado interés en las proteínas vegetales dentro de la industria alimentaria [6]. Los cereales, como fuentes ricas en proteínas vegetales, están ganando protagonismo en el desarrollo de productos que buscan reemplazar o complementar las proteínas animales en la dieta humana. Diversos estudios han demostrado que, con una adecuada combinación y procesamiento, las proteínas derivadas de cultivos como la avena (*Avena sativa* L.), la cebada (*Hordeum vulgare* L.) y el triticale (X *Triticosecale* Wittmack) pueden proporcionar perfiles de aminoácidos que satisfagan los requerimientos nutricionales humanos [7]. Esta tendencia hacia la diversificación de fuentes proteicas no solo responde a preocupaciones de sostenibilidad ambiental, sino también a la búsqueda de opciones alimentarias más saludables y económicamente accesibles [8].

En este contexto, la cuantificación del contenido proteico en cereales constituye un parámetro crítico para evaluar su calidad nutricional, optimizar procesos agroindustriales y desarrollar programas de mejoramiento genético. Los métodos analíticos convencionales para la determinación de proteínas en cereales, como Kjeldahl y Dumas [9], aunque precisos, presentan limitaciones inherentes: destructividad de la muestra, alto consumo de reactivos y tiempos prolongados de análisis, factores incompatibles con los requisitos de procesamiento a gran escala demandados en la agricultura.

La espectroscopía de infrarrojo cercano (NIR sus siglas en inglés) y más tarde las imágenes hiperespectrales (HSI sus siglas en inglés) han emergido como alternativas tecnológicas paradigmáticas, permitiendo análisis rápidos, no destructivos, ambientalmente sostenibles [10] y eficientes para la evaluación del contenido proteico [11, 12, 13, 14, 15, 16]. Recientes avances en instrumentación portátil han ampliado las aplicaciones in situ, al combinar precisión de laboratorio con versatilidad operativa [17]. No obstante, el análisis de granos intactos introduce complejidades espectrales derivadas de efectos de dispersión multicapa, heterogeneidad morfológica y variabilidad en el empaquetamiento celular, lo que requiere estrategias avanzadas de preprocesamiento y modelización [18, 19, 20].

En el campo de procesamiento y modelización, machine learning (ML) permite procesar y analizar grandes y complejos volúmenes de datos, siendo capaces de identificar patrones y relaciones entre ellos, frente a los métodos tradicionales. Recientemente, un estudio destacaba la eficacia de ML en el análisis de cultivos de cereales para mejorar el rendimiento [21].

El trabajo que aquí se presenta analiza los datos obtenidos de proteína por el Instituto de Investigaciones Agrarias Finca La Orden-Valdesequera del Centro de Investigaciones Científicas y Tecnológicas (CICYTEX) de tres especies de cereales (2.2). Se presenta un enfoque basado en Python y desarrollado de forma modular y ampliable, que aplica y compara tres algoritmos bien conocidos de Machine Learning, para predecir la proteína de muestras de cereal a partir de su información espectral basada en NIR. Posteriormente, mediante programación genética se intenta ajustar la predicción de la proteína atendiendo a la relación de las predicciones de los modelos de ML optimizados en la fase anterior, dando como resultado una expresión matemática que relaciona de forma explicable los modelos de ML.

Bajo el paradigma de la Inteligencia Artificial Explicable y la ciencia abierta, este estudio busca determinar el método más adecuado para predecir el contenido proteico en granos intactos de avena, cebada y triticale mediante NIR, empleando algoritmos de aprendizaje automático y programación genética. Para ello, se comparan los distintos enfoques mediante métricas estandarizadas, con el fin de seleccionar la estrategia más precisa.

## 2  Materiales y métodos

Esta sección recoge las metodologías utilizadas en este trabajo, desde el material vegetal, análisis de NIRS, método de referencia de determinación de la proteína, descripción del sistema algorítmico empleado, junto con estrategias de ML y programación genética.

### 2.1  Material vegetal

En este estudio preliminar se emplearon 914 muestras de grano pertenecientes a tres especies diferentes: avena (*Avena sativa*) ($N_{Av}$=186), cebada (*Hordeum vulgare*) ($N_{Ce}$=184) y triticale (X *Triticosecale* Wittmack) ($N_{Tr}$=544). Esta muestras provienen de ensayos realizados por la Red GENVCE (Grupo para la Evaluación de Nuevas Variedades de Cultivos Extensivos en España) durante la campaña 2023-2024. GENVCE es una organización que agrupa a técnicos de diversos centros e institutos de investigación de las comunidades autónomas españolas, dedicados a la evaluación agronómica y de calidad de nuevas variedades de cultivos extensivos en diferentes regiones cerealistas del país. Las muestras analizadas en este trabajo fueron recolectadas de parcelas experimentales ubicadas en distintas localidades de Extremadura (España), representando una variedad de condiciones agroclimáticas. La diversidad genética y ambiental de las muestras, garantizada por la metodología de GENVCE, proporciona una base sólida para el desarrollo y validación de modelos predictivos aplicables en diferentes contextos agronómicos.

### 2.2  Análisis NIRS y métodos de referencia para la determinación de la proteína

Para la determinación del contenido de proteína en los granos de avena (*Avena sativa* L.), cebada (*Hordeum vulgare* L.) y triticale (X *Triticosecale* Wittmack), se utilizó la espectroscopía de infrarrojo cercano como método no destructivo, complementado con el método de referencia de combustión Dumas. A diferencia de los análisis convencionales que emplean muestras molidas para mejorar la homogeneidad y reducir variaciones ópticas, en este estudio se optó por la medición en granos intactos, lo que supone un desafío adicional debido a la heterogeneidad morfológica y estructural de las muestras. Factores como la dureza del grano, la variabilidad en la superficie y la distribución interna de los componentes afectan la dispersión de la luz y, por ende, la calidad espectral, lo que requiere una optimización cuidadosa del análisis espectral y el procesamiento de datos.

Los espectros se obtuvieron mediante un espectrómetro LabSpec 2500 (ASD Inc.®) equipado con un sensor UNIT5065, con una resolución espectral de 2 nm en el rango de 1000 a 2500 nm. Se empleó una sonda de reflectancia difusa ASD® Turntable junto con una fuente de luz halógena para garantizar una iluminación uniforme de las muestras. Cada medición se realizó asegurando que el grano cubriera completamente la superficie de la cápsula de medición, minimizando efectos de dispersión no deseados. Para mejorar la estabilidad y fiabilidad del análisis, se registró un espectro de referencia con una placa cerámica blanca antes de cada conjunto de 25 muestras.

Los espectros obtenidos fueron analizados en bruto, tanto en reflectancia como en absorbancia ($\log_{10}(1/R)$). Posteriormente, se aplicaron distintos preprocesamientos espectrales con el objetivo de reducir el ruido y mejorar la calidad de los datos, mitigando los efectos de dispersión óptica y

permitiendo mejorar la precisión de los modelos predictivos. Entre las técnicas consideradas, se incluyeron correcciones de dispersión de luz, normalización y filtrado de ruido. Como resultado del análisis comparativo, los preprocesamientos que ofrecieron los mejores ajustes en la predicción del contenido proteico fueron la Corrección Multiplicativa de Dispersión (MSC) y la primera derivada de Savitzky-Golay (1D_2-7-7).

- Corrección Multiplicativa de Dispersión (MSC) permitió corregir la dispersión de la luz provocada por diferencias en la superficie y morfología de los granos, reduciendo las variaciones ópticas no relacionadas con la composición química. Al ajustar cada espectro a un modelo de referencia basado en la media del conjunto de datos, se logró minimizar la influencia de irregularidades físicas y optimizar la linealidad de los datos espectrales [22].

- La primera derivada de Savitzky-Golay (1D_2-7-7) consiguió mejorar la resolución de los picos espectrales, eliminar tendencias de fondo y reducir el ruido espectral que afectan la calidad de los espectros, lo que facilita la identificación de las señales relacionadas con el contenido proteico. El filtro de Savitzky-Golay es una técnica de suavizado que ajusta un polinomio de bajo grado (segundo orden en este caso) sobre una ventana de datos móviles (de 7 puntos a cada lado del dato central, es decir, 15 en total), calculando la primera derivada en cada posición. Este tipo de pretratamiento es especialmente útil cuando los espectros presentan señales superpuestas o con interferencias derivadas de la estructura del grano.

Para la cuantificación del contenido proteico, se empleó el método de combustión Dumas mediante un analizador LECO FP-528, ampliamente utilizado para la determinación de nitrógeno total en muestras agroalimentarias. La cantidad de nitrógeno obtenida se convirtió en contenido proteico utilizando un factor de conversión, para la cebada se utilizó 5.88, en avena 5.50 y en triticale 5.78, convencionalmente aceptados para estas especies de cereales.

Dado que los granos de avena, cebada y triticale presentan diferencias estructurales y de composición química, su interacción con la radiación NIR varía en función de la especie, lo que afecta a la respuesta espectral y a la precisión de los modelos predictivos. La evaluación de múltiples preprocesamientos espectrales y la selección de aquellos con mejor desempeño fueron claves para mejorar la robustez y aplicabilidad de los modelos de espectroscopía NIR en la determinación no destructiva del contenido proteico en estos cereales.

Para estudiar la fiabilidad del modelo final presentado, se emplea métricas estandarizadas, priorizando aquellos que minimizaran los errores y maximizaran la capacidad de ajuste. Se consideraron los siguientes parámetros estadísticos:

- Coeficiente de determinación ($R^2$) de entrenamiento y test.
- Error cuadrático medio (MSE) de entrenamiento y test.
- Relación de desempeño de predicción (RPD) (ver expresión (1)). Se considera que un RPD > 2.5 indica buena capacidad predictiva, mientras que valores superiores a 3.0 sugieren un modelo óptimo para aplicaciones industriales [23, 24].
- Rango y Error de Predicción (RER, ver expresión (1)), una métrica menos utilizada pero útil para evaluar la aplicabilidad práctica del modelo en la industria. En general, se considera que un RER superior a 10 indica una buena capacidad predictiva de los modelos NIRS [25, 26].

$$RDP = \frac{\sigma_{predicciones}}{\overline{MSE}} \qquad RER = \frac{\text{Proteína}_{\max} - \text{Proteína}_{\min}}{\overline{MSE}} \qquad (1)$$

## 2.3 Técnicas de Machine Learning

La integración de técnicas de ML para estimar el contenido proteico en cereales ofrece ventajas significativas para trabajar con datos complejos, precisión, eficiencia y adaptabilidad a los problemas, lo que hace del uso de estas técnicas idóneas para el trabajo aquí presentado. Para este trabajo se han utilizado las siguientes técnicas de ML:

- **Regresión de mínimos cuadrados parciales (PLS)**. Método estadístico que permite modelar relaciones entre variables independientes y dependientes que presentan colinealidad o

alta dimensionalidad [27]. Es ampliamente utilizado en espectroscopía porque se maneja bien con datos de alta dimensión y reduce el sobreaprendizaje.

- **Random Forest (RF)**. Método de aprendizaje conjunto que construye múltiples árboles de decisión durante el entrenamiento y la salida es la media de las predicciones [28]. Se utiliza ampliamente por su robustez, precisión y capacidad para manejar grandes volúmenes de datos y de alta dimensión. RF es menos propenso al sobreajuste y funciona bien ante la presencia de valores atípicos o la falta de valores (Missing values), aunque computacionalmente es más costoso.

- **k-Nearest Neighbors (KNN)**. Algoritmo de aprendizaje supervisado no paramétrico, que se basa en la búsqueda de vecinos más cercanos en el espacio de características. Realiza la predicción estimando el valor promedio de los k vecinos más cercanos. Es un algoritmo idóneo para relaciones no lineales, aunque es sensible al valor de k y a la escala de los datos [29].

- **Programación genética (GP)**. La programación genética se basa en la selección natural para hacer evolucionar funciones computacionales [30]. GP no necesita conocimiento previo sobre la solución, sólo el mecanismo para determinar qué solución es mejor que otra.

## 2.4 Sistema algorítmico

Tal y como se ha detallado en la subsección anterior, se han utilizado algoritmos de ML ampliamente conocidos, que se adaptan de manera efectiva a las condiciones del problema abordado en este trabajo. Si bien el uso de algoritmos de ML es ampliamente reconocido, la complejidad del presente estudio aumenta debido a la alta dimensionalidad de los datos de entrada: se trabaja con 1500 variables dependientes, correspondientes a los valores de longitud de onda expresados en $nms$ que caracterizan cada muestra. La selección de estos algoritmos de ML responde a su reconocida adaptabilidad a este tipo de problemas.

El sistema de predicción presentado en este trabajo puede dividirse en dos subsistemas, donde en una primera fase se optimizan los algoritmos de ML indicados. Para ello se proporciona el dataset, y se divide en un conjunto de entrenamiento y test utilizando la técnica de validación cruzada 5-fold cross-validation. Los conjuntos de entrenamiento y test resultantes son utilizados por los diferentes algoritmos de ML para su optimización, ajustando sus hiperparámetros con la conocida técnica de GridSearchCV, adatpada a cada uno de los algoritmos utilizados, RF, PLS y KNN.

Una vez los algoritmos de ML han sido optimizados y los modelos resultantes almacenados, se hace uso de estos para generar un nuevo dataset formado por 4 columnas. Tres de ellas serán las variables dependientes, correspondientes a la predicción ofrecida por cada uno de los algoritmos de ML optimizados en la fase anterior. La cuarta columna pertenece a la variable independiente, que para este caso nuevamente es la proteína. Se genera este nuevo dataset, del mismo tamaño que el dataset original, ya que en esta segunda fase se pretende diseñar una función dependiente de las tres predicciones arrojadas por los algoritmos de ML para poder predecir de forma más ajustada la variable objetivo de este trabajo, la proteína. Para esta nueva optimización se ha utilizado la técnica de la programación genética, ya que es ampliamente conocida su capacidad de diseñar funciones dependientes de variables de entrada y poder determinar valores de salida, encontrando una relación matemática entre ellas. Lo cual nos permite poder explicar el modelo final obtenido, gracias a la interpretabilidad de las soluciones aportadas. Para poder optimizar de forma correcta la segunda fase donde se utliza el algorirmo basados en PG, se divide el nuevo conjunto utilizando nuevamente 5-fold cross-validation, generando un nuevo conjunto de entrenamiento y test, necesario en el proceso de optimización y validación del algoritmo basado en PG.

La figura 1 muestra un ejemplo de funcionamiento del sistema presentado en este trabajo.

## 3 Resultados

Atendiendo al esquema presentado en la figura 1, podemos entender que el funcionamiento se divide en dos fases. En primer lugar, se han optimizado de forma independiente los algoritmos de ML, utilizando para ello el optimizador conocido como GridSearchCV. Las tablas 1 y 2 muestran los valores de los hiperparámetros del mejor modelo encontrado para cada uno de los algoritmos utilizados, RF, PLS y KNN.

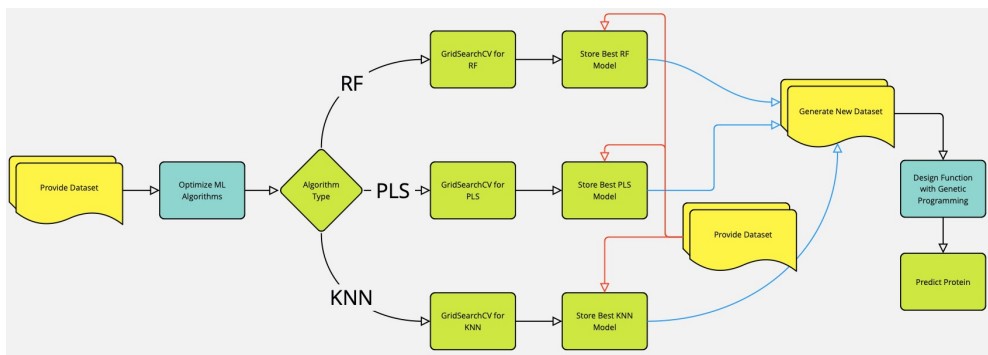

Figure 1: Esquema de funcionamiento.

Table 1: Parámetros de los modelos para conjunto de datos `MSC`

| Modelo | Hiper-parámetros | | | | |
|--------|------------------|------------------|------------------|------------------|------------------|
| KNN | leaf_size 10 | metric minkowski | n_neighbors 3 | p 1 | weights distance |
| RF | max_depth 10 | min_samples_leaf 1 | min_samples_split 2 | n_estimators 100 | |
| PLS | max_iter 500 | n_components 15 | scale True | tol 0,0001 | |

Las tablas 3 y 4 muestran los valores de MSE y $R^2$ para cada uno de los tratamientos y modelos utilizados. Basándonos en los resultados promedios alcanzados sobre el conjunto de test, se observa que los modelos con mejores predicciones son aquellos que se han optimizado utilizando el preprocesamiento espectral MSC.

Una vez ajustados los diferentes algoritmos de ML, se desarrolla un nuevo modelo basado en programación genética. El objetivo de este modelo es buscar una relación entre las predicciones obtenidas con los modelos de ML y un mejor ajuste en la predicción final de la proteína. Para ello, se genera un nuevo conjunto de datos, donde se reflejan las predicciones de los modelos de ML junto con la variable dependiente, *Proteína*. Este conjunto de datos se divide en 5 conjuntos, utilizando validacion cruzada 5-fold cross-validation. Esto nos permite poder realizar varias optimizaciones del nuevo modelo basado en GP, utilizando conjuntos de entrenamiento y test diferentes para determinar la robustez del modelo.

Los operadores aritméticos utilizados por el algoritmo de GP son los utilizados de forma estándar en regresión simbólica, tales como suma, resta, multiplicación, división protegida, negación, raíz cuadrada protegida, logaritmo protegido, exponencial protegida, seno y coseno, junto con una constante. Indicar la necesidad de operaciones protegidas para no incurrir en operaciones matemáticas inválidas del tipo división por cero o raíces cuadradas de números negativos. El método de selección es torneo con 3 individuos, operador de cruce en un punto, operador de mutación uniforme, con

Table 2: Parámetros de los modelos para conjunto de datos `1D_2-7-7`

| Modelo | Hiper-parámetros | | | | |
|--------|------------------|------------------|------------------|------------------|------------------|
| KNN | leaf_size 10 | metric minkowski | n_neighbors 3 | p 1 | weights distance |
| RF | max_depth 10 | min_samples_leaf 1 | min_samples_split 5 | n_estimators 100 | |
| PLS | max_iter 500 | n_components 20 | scale False | tol 0,0001 | |

Table 3: Resultados de los modelos utlizando el conjunto de datos MSC

| Modelo | $R^2$ (Training) | MSE (Training) | $R^2$ (Test) | MSE (Test) |
|---|---|---|---|---|
| PLS | 0,9231 | 0,2708 | 0,8430 | 0,5837 |
| RF | 0,9704 | 0,1041 | 0,7950 | 0,7623 |
| KNN | 1,0000 | 0,0000 | 0,8453 | 0,5752 |

Table 4: Resultados de los modelos utlizando el conjunto de datos 1D_2-7-7

| Modelo | $R^2$ (Training) | MSE (Training) | $R^2$ (Test) | MSE (Test) |
|---|---|---|---|---|
| PLS | 0,9115 | 0,3206 | 0,8296 | 0,5727 |
| RF | 0,9753 | 0,0896 | 0.8438 | 0,5252 |
| KNN | 1,0000 | 0,0000 | 0.6473 | 1,1857 |

tasa de cruce de $0, 5$, tasa de mutación de $0, 2$, 100 individuos por población y 20 generaciones. La función fitness se basa en el minimizar el MSE.

Table 5: Comparación de resultados para 1D_2-7-7 vs. MSC.

| Ejecución | Tratamiento 1D_2-7-7 | | | | Tratamiento MSC | | | |
|---|---|---|---|---|---|---|---|---|
| | train_mse | test_mse | r2_train | r2_test | train_mse | test_mse | r2_train | r2_test |
| 1 | 0,1766 | 0,1777 | 0,9491 | 0,9547 | 0,1152 | 0,1098 | 0,9671 | 0,9708 |
| 2 | 0,1766 | 0,1777 | 0,9491 | 0,9547 | 0,1163 | 0,1106 | 0,9668 | 0,9706 |
| 3 | 0,1333 | 0,1980 | 0,9616 | 0,9496 | 0,1162 | 0,1107 | 0,9669 | 0,9706 |
| 4 | 0,1333 | 0,1980 | 0,9616 | 0,9496 | 0,1163 | 0,1106 | 0,9668 | 0,9706 |
| 5 | 0,1333 | 0,1980 | 0,9616 | 0,9496 | 0,1000 | 0,1030 | 0,9715 | 0,9727 |
| 6 | 0,1749 | 0,1783 | 0,9496 | 0,9546 | 0,1163 | 0,1110 | 0,9668 | 0,9705 |
| 7 | 0,1849 | 0,1446 | 0,9469 | 0,9629 | 0,1204 | 0,0940 | 0,9676 | 0,9680 |
| 8 | 0,1535 | 0,1150 | 0,9559 | 0,9705 | 0,1204 | 0,0940 | 0,9676 | 0,9680 |
| 9 | 0,1849 | 0,1446 | 0,9469 | 0,9629 | 0,1204 | 0,0940 | 0,9676 | 0,9680 |
| 10 | 0,1540 | 0,1153 | 0,9557 | 0,9704 | 0,1093 | 0,0838 | 0,9706 | 0,9714 |
| 11 | 0,1540 | 0,1153 | 0,9557 | 0,9704 | 0,1204 | 0,0940 | 0,9676 | 0,9680 |
| 12 | 0,1680 | 0,1421 | 0,9517 | 0,9636 | 0,1204 | 0,0940 | 0,9676 | 0,9680 |
| 13 | 0,1764 | 0,1787 | 0,9516 | 0,9445 | 0,1318 | 0,0485 | 0,9630 | 0,9862 |
| 14 | 0,1764 | 0,1787 | 0,9516 | 0,9445 | 0,1318 | 0,0485 | 0,9630 | 0,9862 |
| 15 | 0,1489 | 0,1360 | 0,9591 | 0,9578 | 0,1318 | 0,0485 | 0,9630 | 0,9862 |
| 16 | 0,1489 | 0,1360 | 0,9591 | 0,9578 | 0,1147 | 0,0452 | 0,9678 | 0,9872 |
| 17 | 0,1489 | 0,1360 | 0,9591 | 0,9578 | 0,1318 | 0,0485 | 0,9630 | 0,9862 |
| 18 | 0,1764 | 0,1787 | 0,9516 | 0,9445 | 0,1312 | 0,0482 | 0,9632 | **0,9863** |
| 19 | 0,1858 | 0,1409 | 0,9489 | 0,9570 | 0,0880 | 0,2235 | 0,9735 | 0,9481 |
| 20 | 0,1763 | 0,1355 | 0,9515 | 0,9587 | 0,0880 | 0,2235 | 0,9735 | 0,9481 |
| 21 | 0,1579 | 0,0883 | 0,9566 | **0,9731** | 0,0880 | 0,2235 | 0,9735 | 0,9481 |
| 22 | 0,1858 | 0,1409 | 0,9489 | 0,9570 | 0,0880 | 0,2235 | 0,9735 | 0,9481 |
| 23 | 0,1606 | 0,0886 | 0,9558 | 0,9729 | 0,0880 | 0,2235 | 0,9735 | 0,9481 |
| 24 | 0,1858 | 0,1409 | 0,9489 | 0,9570 | 0,0880 | 0,2235 | 0,9735 | 0,9481 |
| 25 | 0,1345 | 0,1934 | 0,9625 | 0,9440 | 0,1191 | 0,0990 | 0,9677 | 0,9669 |
| 26 | 0,1593 | 0,2419 | 0,9557 | 0,9300 | 0,1191 | 0,0990 | 0,9677 | 0,9669 |
| 27 | 0,1422 | 0,2036 | 0,9604 | 0,9411 | 0,1191 | 0,0990 | 0,9677 | 0,9669 |
| 28 | 0,1605 | 0,2425 | 0,9553 | 0,9298 | 0,1191 | 0,0990 | 0,9677 | 0,9669 |
| 29 | 0,1345 | 0,1934 | 0,9625 | 0,9440 | 0,1191 | 0,0990 | 0,9677 | 0,9669 |
| 30 | 0,1499 | 0,2418 | 0,9583 | 0,9300 | 0,1191 | 0,0990 | 0,9677 | 0,9669 |

Según los datos presentados en la tabla 5, se puede observar que en la ejecución 18 se obtiene el modelo basado en GP con mayor $R^2$ en test, utilizando el pretratameinto espectral MSC. La siguiente expresión es la generada por el modelo en esa ejecución:

$$\text{Sea } A = e^{0,9525969962420688}, \quad D = e^{e^{0,10196347193424526}}.$$

Entonces, la expresión es: $\text{Exp} = \text{Pred\_KNN} - \dfrac{\text{Pred\_KNN} - \text{Pred\_PLS}}{(A + \text{Pred\_KNN}) \times (A + \text{Pred\_KNN} + D)}$ .

Análogamente, utilizando el pretratamiento 1D_2-7-7, se alcanza el mejor modelo en la ejecución 21, donde se puede observar el mejor valor de $R^2$ en test, siendo el mejor individuo en esta ejecución el determinado por la expresión siguiente.

$$\text{Exp} = \sqrt{(\text{Pred\_KNN} \times \text{Pred\_RF}) - (\text{Pred\_KNN} - \text{Pred\_RF})}.$$

Para determinar la robustez de los modelos basados en GP, la tabla 6 muestra los datos de media ($\overline{x}$) y desviación típica ($\sigma$) para el conjunto de ejecuciones realizadas, pudiendo observar una desviación típica muy baja, lo que nos indica la robustez de los modelos optimizados.

Table 6: Resultados estadísticos por tratamiento

| Tratamiento | train_mse | test_mse | r2_train | r2_test | Descripción |
|---|---|---|---|---|---|
| 1D_2-7-7 | 0,1613 | 0,1634 | 0,9548 | 0,9539 | $\overline{x}$ |
| | 0,0181 | 0.0419 | 0,0050 | 0,0122 | $\sigma$ |
| MSC | 0,1136 | 0.1144 | 0,9682 | 0,9682 | $\overline{x}$ |
| | 0,0146 | 0.0595 | 0,0034 | 0,0124 | $\sigma$ |

Finalmente, la fiabilidad de los modelos de ML y del modelo final basado en GP se estudia mediante los valores de RDP y RER presentados en la tabla 7, la cual nos indica que los modelos son óptimos para aplicaciones industriales, y tienen una buena capacidad predictiva, destacando el modelo basado en GP sobre los demás, nuevamente, confirmando que el preprocesamiento espectral basado en MSC obtiene mejores resultados que el tratamiento 1D_2-7-7.

## 4  Conclusiones

Este trabajo confirma la alta fiabilidad de la espectroscopía de infrarrojo cercano para la determinación rápida y no destructiva del contenido proteico en cereales como la avena, la cebada y el triticale. El uso de métodos de preprocesamiento espectral adecuados, como la Corrección Multiplicativa de Dispersión y la primera derivada de Savitzky-Golay (2-7-7), logran disminuir significativamente los problemas derivados de la heterogeneidad del grano intacto y mejoran notablemente la precisión de los modelos de predicción. Gracias a estas técnicas, se superan las limitaciones de los métodos convencionales y se maximizan las ventajas de los sistemas de espectroscopía para uso en la industria agroalimentaria.

Asimismo, la combinación de modelos de Machine Learning (KNN, Random Forest y PLS) aporta versatilidad y robustez al proceso predictivo. Cada uno aborda la complejidad de los espectros de forma complementaria y, al unirse con programación genética, se obtiene una función integradora capaz de explicar y mejorar la estimación final de la proteína en los cereales. Este enfoque híbrido, convalidado mediante validación cruzada y métricas estandarizadas, confirma el gran potencial de la Inteligencia Artificial para optimizar análisis masivos de muestras de grano en entornos de producción a gran escala.

El trabajo aquí presentado aporta un amplio nivel de interpretabilidad, ya que la programación genética no solo realiza predicciones precisas, sino que también brinda una expresión matemática

Table 7: Valores de RDP y RER para los algoritmos en los tratamientos MSC y 1D_2-7-7

| Algoritmo | MSC | | 1D_2-7-7 | |
|---|---|---|---|---|
| | RDP | RER | RDP | RER |
| RF | 2.5367 | 11.5571 | 3.6820 | 16.7746 |
| PLS | 3.3129 | 15.0934 | 3.3766 | 15.3833 |
| KNN | 3.3619 | 15.3164 | 1.6310 | 7.4308 |
| GP | 18.8938 | 89.0501 | 12.1464 | 56.6939 |

que describe la relación entre los diferentes modelos y el contenido proteico. Esta cualidad explicable refuerza la comprensión científica y abre la puerta a futuras mejoras.

En conclusión, el trabajo aquí presentado demuestra un sistema de predicción sólido y escalable, con amplias posibilidades de adaptación a otras especies de cereales o incluso a diferentes tipos de análisis de calidad agrícola. La combinación de NIRS, preprocesamientos espectrales avanzados y algoritmos de Machine Learning de última generación, consolida un marco eficaz que puede reducir el tiempo y los costos de la industria agroalimentaria, impulsando el desarrollo de soluciones cada vez más precisas y sostenibles.

## Agradecimientos

Este trabajo está financiado por el Ministerio Español de Ciencia e Innovación con el proyecto PID2023-147409NB-C22 y MCIN/AEI/10.-13039/501100011033 y Cátedra de Ciberseguridad INCIBE-UEx-CUMe (C110/23), fruto del convenio de colaboración suscrito entre el Instituto Nacional de Ciberseguridad (INCIBE) y la Universidad de Extremadura. Esta iniciativa se realiza en el marco de los fondos del Plan de Recuperación, Transformación y Resiliencia, financiados por la Unión Europea (Next Generation)

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
