# OpenReview forum: "Enfoque integrado de NIRS, Machine Learning y programación  genética para la estimación explicable del contenido proteico en cereales"
_MAEB/2025/Congreso — MAEB 2025_

### Official Review · Reviewer_Mp2z · 2025-03-18
**Este trabajo propone la combinación de tres métodos de regresión con el uso de programación genética para para la estimación del contenido proteico en cereales a partir de datos de espectroscopía NIRS.**

**Rating:** 2
**Confidence:** 5

**Review:**

El objetivo del  trabajo es relevante en tanto una predicción precisa basada en datos NIRS evitaría la destrucción de los granos para predecir su contenido proteíco.

En general el trabajo es claro respecto a la presentación del problema y de la estrategia utilizada. Sin embargo, el trabajo presenta un número de limitaciones y posibles errores metodológicos, no ofrece soluciones originales y no es concluyente respecto a las ventajas del método introducido.

1) La utilización de las predicciones  de clasificadores o regresores de un primer nivel como entrada para el aprendizaje de un modelo en un segundo nivel es la base de los métodos de stacking. En la literatura de ML existen múltiples ejemplos de la conveniencia de este tipo de aproximaciones. Los autores deben recalcar en qué consiste la novedad del trabajo desde el punto de vista metodológico.

2) El valor explicativo del modelo de segundo nivel parece bastante reducido. En general la explicabilidad suele definirse con respecto a las instancias o ejemplos de entrada. Un método de explicabilidad debería indicarnos qué elementos o características de los ejemplos determinan la predicción dada por el modelo en cuestión. En este trabajo el resultado de modelo GP hace referencia a las salidas de los tres primeros modelos base. Es conocido que el si bien los árboles de decisión pueden ser fácilmente interpretables el Random Forest no es un modelo fácilmente explicable, particularmente si incluye numerosos árboles de decisión. Defender la "explicabilidad" de la aproximación por el hecho de conocer cómo se combinan los modelos base es un argumento débil.

3) Probablemente la carencia mayor del trabajo está en el proceso de validación de la combinación de los modelos base y GP. En el mejor de los casos, la información disponible en el trabajo sobre este proceso de validación es confusa y escasa. En el peor de los casos la validación del segundo nivel es incorrecta y los resultados obtenidos parecen indicar valores de precisión optimistas. En el caso del aprendizaje  y validación de los tres clasificadores bases usando 5-fold crossvalidation la validación es correcta. Sin embargo, a la hora de aprender la combinación de modelos usando GP no queda claro si se ha mantenido la misma partición de los datos, aprendiendo un modelo GP dentro de un enfoque de crossvalidación, es decir, realizando 5x30 ejecuciones en cada caso o si se han dividido los datos en dos conjuntos train+test, o si se ha utilizado otro conjunto de datos no usado inicialmente. Si se han utilizado los mismos datos iniciales (con las predicciones generadas por los tres modelos) es muy probable que se esté dando un caso de data-leaking, incluso si los datos originales se han divido en dos grupos train+test.
4) La explicación de GP dada en la pág 5, línea 189 debe ser un poco más precisa y rigurosa.
5) En las tablas 1 y 2, no queda claro el sentido de "weights--distance"
6) La presentación de datos en la Tabla 5 no es muy útil para mostrar el comportamiento del algoritmo. Sería mejor presentar mean, best, std de las 30 ejecuciones para cada una de las métricas en lugar de mostrar los resultados de cada una. Resulta significativo y sospechoso que para semillas consecutivas (20-24) y (25-30) los resultados para la base de datos MSC sean idénticos.
7) Para evaluar la validez de GP convendría comparar con otro enfoque de stacking clásico en el cual el modelo de segundo nivel es más simple, por ejemplo un modelo de regresión a partir de los tres inputs.
8) Pag. 8, línea 275. Afirmar que el enfoque propuesto "aporta un amplio nivel de explicabilidad" no parece corresponderse a la realidad. Conocer la manera en que cada modelo influye en la predicción no es necesariamente explicabilidad.




Otros comentarios

- Eq. (1), no se explica el significado de sigma
- En la línea 68, pag. la mención a 2.2 parece ser un typo
- Se introduce varias veces en el texto el mismo acrónimo NIR

---

### Official Review · Reviewer_5ZaA · 2025-03-19
**Interesante problema, pero falta mejorar la explicación**

**Rating:** 4
**Confidence:** 4

**Review:**

En este trabajo los autores abordan el problema de la estimación proteica de cereales cultivados. El enfoque que se sigue es la técnica de Espectroscopia de Infrarojo que plantea una serie de retos computacionales que necesita ser abordados. En ese sentido, los autores emplean una serie de algoritmos de aprendizaje automático y de programación genética. El problemas es muy interesante, pero falta explicar mejor el pipeline que se sigue.

Puntos fuertes
===========
- El problema que se aborda en el paper es interesante, relevante, actual y de una complejidad notable. He disfrutado de la lectura. No solo eso sino que además utilizan datos reales y actuales de parcelas experimentales. Doy la enhorabuena a los autores.

Puntos débiles
===========

- Uno de los problemas que veo tanto en la redacción del abstract y la introducción está en el entendimiento del pipeline llevado a cabo. En la línea 58 de la página 2, los autores tratan de explicar el motivo por el cual los datos proporcionados por el NIR necesitan tratamiento y suponen un reto. No obstante, la mayoría de los lectores desconocer la técnica NIR (como es mi caso), y por lo tanto, resulta complicado entender la necesidad de las técnicas de aprendizaje automático. ¿Qué es lo que devuelve el NIR? ¿son imágenes? ¿Cual es la tarea de predicción en las mismas?

- Siguiendo con el anterior punto, algo semejante ocurre con la Programación genética. ¿Sobre que contexto de van a aplicar? ¿Por qué las expresiones matemáticas permitirán mejorar la estimación de la proteína en los cereales? ¿cómo? Es cierto que todo ello se entiende una vez los capítulos avanzan, pero el lector no llega a comprender el alcance del trabajo leyendo el abstract o la introducción.

- Otro de los problemas fundamentales que veo está en entender el “sistema algorítmico”, y más concretamente el párrafo 1 de la sección 2.4. Lo que pasa es que el lector desconoce totalmente el output del NIR, ni como es la variable a predecir. Estamos hablando de ¿clasificación? ¿regresión? Hay una ausencia total del “learning scenario”, y creo que en este trabajo resulta fundamental.


- Typos:
	+ línea 100, espectroscopia.
	+ machine learning -> aprendizaje automático.
	+ línea 68, falta paréntesis.
	+ el término pretratamiento suena extraño, quizá sea mejor emplear la palabra “preprocesamiento”.
	+ línea 178, falta paréntesis.
	+ hay algunas más, please revisad bien el documento.

---

### Official Review · Reviewer_iheV · 2025-03-19
**No está clara la contribucion realizada, ni su relevancia en el contexto de la programacion genética.**

**Rating:** 2
**Confidence:** 5

**Review:**

Este estudio propone un enfoque basado en espectroscopía de infrarrojo cercano (NIRS), aprendizaje automático y programación genética para predecir el contenido proteico en cereales de manera explicable. El tema es interesante y se puede considerar dentro del ámbito del congreso. Sin embargo, despues de una primera lectura, tengo algunas críticas que hacer al trabajo presentado.

En primer lugar, la redaccion no está muy alineada con la forma de presentar los trabajos en MAEB. El principal objetivo de este congreso es el desarrollo de metaheurísticas, algoritmos evolutivos y bioinspirados y, aunque se toca el tema de la PG, la importancia de esta parte en el trabajo es bastante cuestionable.

Las partes introductorias son demasiado verbosas. Creo que la introduccion es demasiado extensa y que aporta bastante poco a la contribucion que se desea vender.

La seccion 2 detalla muchos aspectos de la obtencion de los datos a través de las muestras, pero no de la algorítmica que integra el sistema desarrollado. Me sorprende muchísimo que no haya mención ni explicacion a la PG desarrollada.

El análisis de hiperparámetros se comenta en la seccion 3 (resultados). En esta seccion se dedica un párrafo a describir muy superficialmente los nodos terminal/función del sistema de GP. Tal como se ha redactado el trabajo, es imposible replicarlo. En mi opinion, esto es un defecto grave del trabajo.

La Tabla 5 no parece aportar demasiado.

En conclusión. No queda clara la contribucion al campo del a computación evolutiva, ni la ventaja de usar como parte final del procedimiento programación genética. Hace referencia a ecuaciones bastante sencillas, lo cual es interesante, pero el procedimiento aparece muy opaco.

---

### Decision · Program_Chairs · 2025-03-20

Accept